# The Multifaceted Function of Water Hyacinth in Maintaining Environmental Sustainability and the Underlying Mechanisms: A Mini Review

**DOI:** 10.3390/ijerph192416725

**Published:** 2022-12-13

**Authors:** Jing Xu, Xiaoxiao Li, Tianpeng Gao

**Affiliations:** Key Laboratory for Ecological Restoration and High Quality Development of Qinling Mountains in the Upper and Middle Reaches of Yellow River, School of Biological and Environmental Engineering, Academy of Eco Xi’an, Xi’an University, Xi’an 710065, China

**Keywords:** water hyacinth, environmental sustainability, pollutants remediation, green energy, carbon sink

## Abstract

Water hyacinth (*Eichhornia crassipes*) (WH) is a widespread aquatic plant. As a top invasive macrophyte, WH causes enormous economic and ecological losses. To control it, various physical, chemical and biological methods have been developed. However, multiple drawbacks of these methods limited their application. While being a noxious macrophyte, WH has great potential in many areas, such as phytoremediation, manufacture of value-added products, and so on. Resource utilization of WH has enormous benefits and therefore, is a sustainable strategy for its control. In accordance with the increasing urgency of maintaining environmental sustainability, this review concisely introduced up to date WH utilization specifically in pollution remediation and curbing the global warming crisis and discussed the underlying mechanisms.

## 1. Introduction

Water hyacinth (*Eichhornia crassipes*) (WH) is a free-floating perennial aquatic plant, belonging to the monocotyledon *Ponteneriaceae*. WH originates from the amazon basin in South America, and has been successfully introduced to North America, Africa, Asia, Oceania, and Europe, intentionally and unintentionally [1,2]. Moreover, WH is going to invade even more areas due to global climate change [3]. WH is a notorious invasive weed, causing severe ecological problems and great economic losses. The successful invasion of WH is mostly due to its high rate of growth and clonal reproduction, lack of natural enemies, flexibility in growth requirements, and allelopathic effects on other organisms. Once it occupies water surfaces, WH threatens the survival of other organisms by blocking sunshine, consuming oxygen and nutrition in water. In addition, the massive mats of WH provide shelters for pests, such as mosquitos, and thereby increase potential health risks for nearby communities. More detailed information about the ecological problems caused by WH can be found in these reviews [4,5,6]. Moreover, overgrowth of WH hampers irrigation and navigation. To control WH, several strategies have been employed, including physical control by manual removal and mechanical dredging, chemical control by spraying herbicides, biological control by releasing insect enemies and pathogenic fungi, and integrated control by combining two or more of the above-mentioned methods [6]. In addition to the drawbacks, including laborious, environmentally unfriendly and unstable in terms of efficiency, the cost for WH management is relatively high [4,7,8]. For example, almost $46 million has been spent on managing aquatic invasive weeds, mainly WH, in the California Bay-Delta during 2013–2016 [9].

While being a top noxious weed, WH, surprisingly, exhibits considerable potential in many areas. First of all, WH has been widely cultured to feed fish and livestock for its nutrition [10]. In addition to providing nutrition, feeding with WH could enhance the disease resistance of animals because of its antimicrobial activity [11]. In addition, WH extracts showed growth inhibition to a wide range of micro-organisms, including bacteria, fungi and algae to different extents [12,13,14,15]. Moreover, WH extracts exerted anti–cancer/anti-tumor, anti-inflammatory, neuropharmacological, hepatoprotective and wounding healing activities. Therefore, it has long been serving as a medicinal plant in some African and Asian countries [16,17,18,19]. The diverse applications of WH in medical care are mainly attributed to its various secondary metabolites [13]. WH is also an excellent source of planting matrix and fertilizer resulting from its high level of cellulose [20]. Composting with other materials, such as food residues, livestock manure and wood chips, is another common application of WH as a result of its rich cellulose [21]. Richness in cellulose also enables WH to be a decent raw material in producing some other value-added products, such as handicrafts, paper, filter membranes, furfural, fibers and so on [22,23,24,25]. Interestingly, WH can be used in the construction industry. The crude extract of WH that is rich in cellulose, saturated and unsaturated fatty acids was shown to slow down the hydraulic speed and increase the hardening time of the cement, thereby enabling cement to be paved for longer [26]. These applications bring enormous benefits. According to the model proposed by Harun et al. (2021), 1 hectare WH could yield 3500 kg of fresh plants, and these plants can be transformed into 3500 m^3^ biogas or other valuable products at reasonable amounts [27]. Moreover, after purifying water, the WH plants harvested from Dianchi Lake, a 309 km^2^ lake located in China, are supposed to bring a net economic gain of 11.72 million Yuan (RMB) over a 15-year time horizon by providing the harvested WH as the feedstock of biogas production [28]. Compared with the routine landfill option, the biogas option not only brings economic benefits but also reduces greenhouse gas (GHG) emissions. Therefore, resource utilization seems to be a promising and sustainable strategy to control WH.

Although some reviews on WH utilization have already been published, they mainly focused on phytoremediation towards traditional pollutants, such as heavy metals and dyes, or on biofuel production. However, remediation of newly developed pharmaceutical residues and persistent industry contaminants by WH, as well as other applications of WH in curbing the global warming crisis, were barely reviewed. Furthermore, while various utilizations of WH have been well classified, the underlying mechanisms still need to be systematically discussed. Therefore, in addition to summarizing recent progress in phytoremediation of traditional pollutants and in biofuel production, this review sheds light on WH utilization in remediation of new types of pollutants, in generating other types of green energy and in carbon capture, as well as new application forms. Moreover, this work deciphers the mechanisms underlying various applications. Overall, this work aims to provide a concise and in-depth review of WH utilization in maintaining environmental sustainability.

## 2. Applications of WH in Maintaining Environmental Sustainability

### 2.1. Super Adsorbent toward Various Pollutants

WH is a widely used biosorbent in the phytoremediation of various pollutants. Compared with other aquatic plants that are also often used in phytoremediation, such as water lettuce, WH presents superiority in terms of growth output and adsorption efficiency towards some pollutants [29].

#### 2.1.1. Metals

WH has been repetitively proven to profoundly absorb a wide range of heavy metals, light metals and radioactive metals, including Ag, Al, Au, Cd, Co, Cr, Cu, Eu, Fe, Hg, Mn, Ni, Pb, Sb, Sn, Sr, Ti, U, V, Zn [5,30,31,32] (Table 1). The range is still increasing. Recently, Kartamihardja et al. (2021) showed WH effectively removed Gd [33]. The prominent performance of WH in metal absorption enables it to be a bio-indicator for water phytoremediation [34]. Analysis of metal concentrations in different tissues revealed that roots normally contain the highest amount of metals, followed by leaves, and finally the bulbous petioles [5,35,36]. This general rule suggests roots are the main functional tissue of WH in phytoremediation, while the aerial tissues normally contain much less metals and, therefore, are safe enough to feed animals. The metals enriched by WH can be recycled by extraction via physical or chemical methods after plant harvesting and subsequently drying or burning into ashes [30].

#### 2.1.2. Other Inorganic Pollutants

Inorganic salts, including As, N, P, S, and Se, are also the adsorbing targets of WH (Table 1) [5]. Therefore, WH is often used in the phytoremediation of industrial waste water and other water bodies contaminated by these salts [72]. However, it should be paid special attention that WH usually accelerates its growth and clonal reproduction in eutrophic water and thus has a high risk of population outbreaks resulting in quick occupation of water surfaces [73]. In this case, methods, such as the use of floating fences to prevent its spread and image-processing technologies to monitor its population, are necessary to avoid potential secondary ecological problems [74,75].

#### 2.1.3. Dyes

Industrial waste water, such as textile factories, often contains various organic dyes. In addition to changing the pH of water and bringing suspended particles as well as heavy metal pollutions, these dyes also increase the biological oxygen demand, causing ecological deterioration. The most economical and effective solution to this problem at present is phytoremediation by aquatic plants, whereby WH is a commonly used plant. The well-developed floating roots can intercept suspended particles on one hand, and directly absorb dyes in water on the other [40]. Some comprehensive reviews have summarized the species of dyes that could be absorbed by WH, while several recent articles have discovered more [2,37,38,39,40,41,42]. In brief, more than 36 kinds of dyes have been reported to be absorbed by WH, including acid yellow 17, alkaline auramine O, azharanth dye, BF-4B red active dye, black B, BreActive magenta B, C. I. acid blue 19, C. I. acid blue 25, C. I. acid blue 80, C. I. acid green 27, C. I. acid red 1, C. I. acid red 40, CI direct blue 201, cibacron blue FR, cibanone gold yellow RK, congo red, cotton blue B2G, cotton red B2G, cotton yellow 2RFL, cresol red, crystal violet, indosol dark-blue GL, malachite green, methylene blue, methyl red, methyl orange, moxilon blue GRL, phenol red, reactive black 5, reactive blue 21, reactive turquoise blue, red RB, rhodamine B, rose bengal, vat green FFB, and xylenol orange (Table 1). In general, the remediation efficiency of cationic dyes by WH is higher than that of anionic dyes, and the concentration of enriched dyes in roots is higher than that in aerial tissues [38]. Interestingly, most work regarding dye remediation was performed on WH in the form of detached tissues, ash and biomass, rather than intact living plants. Moreover, WH in these forms showed comparative or even higher adsorption capacity, which is determined by parameters including pH, initial concentration of dye, adsorbent dose, and contact time [39].

#### 2.1.4. Pesticides

The application of pesticides, such as insecticides, bactericides and herbicides is indispensable to modern agriculture. Although they greatly reduce yield losses caused by various pathogens and pests, their toxicity on non-target creatures results in severe ecological problems. WH has been reported to absorb a variety of pesticides, including organic chlorine pesticides α–benzene hexachloride (BHC) and its isomers β-BHC, δ-BHC and γ-BHC, β-endosulfan, clomazone, cyhalothrin, dichlorodiphenyltrichloroethane, dicofol and heptachlor epoxide, organophosphorus pesticides, chlorpyrifos, triazine, diazinon, ethoprophos, malathion, methylparathion, omethoate, ethion, as well as herbicides diphenamid, mesotrione and fomesafen [43,44,45,46,47,48]. Generally, the bioaccumulation factors are greater for organic chlorine pesticides than for organophosphorus pesticides. In addition, WH roots accumulate higher levels of pesticides than the aerial part [45]. Most of the above pesticides have been shown to degrade within WH plants, and the degradation efficiency is related to WH root-associated micro-organisms [5,76] (More in Section 3.2.2).

#### 2.1.5. Pharmaceutical Residues

Sulfinirazine is a low-cost and wide-spectrum sulfa antibiotic that is widely used in animal husbandry, and often residues in agricultural wastewater. It has been shown that sulfadiazine is absorbed by WH roots and subsequently transported to the upper tissues by steaming, and it is degraded during this process [49]. In addition, the adsorption and degradation of ciprofloxacin by WH has also been reported [50]. Moreover, WH can efficiently adsorb tetracycline, and interestingly, the addition of Cu^2+^ into water improves the absorption efficiency of tetracycline [77,78]. This work simulates a real environmental scenario whereby multiple types of pollution exist simultaneously and examines the capacity of WH to tackle mixed antibiotic and heavy metal pollution, the results of which turned out to be promising and intriguing.

In addition to antibiotics, WH is capable of remediating other drug pollution. WH was reported to have efficiently absorbed three anti-inflammatory drugs, naproxen, ibuprofen, and diclofenac by roots, and subsequently transported them to the petioles and reached the maximum cumulative concentration in the leaves [51]. Interestingly, when ibuprofen and caffeine co-existed in water, the adsorption efficiency of ibuprofen was significantly improved, although the underlying mechanism remained obscure [79]. Moreover, another non-steroid anti-inflammatory drug, fenoprofen, was detected in WH plants grown in rivers that had been previously found to be contaminated by non-steroidal anti-inflammatory drugs, suggesting that WH is a promising plant for the remediation of this drug [52]. Furthermore, three anti-retrovirus drugs, emtricitabine, tenofovir disoproxil and efavirenz, were detected in WH plants collected near the outfall of wastewater treatment plants [53].

#### 2.1.6. Newly Developed Persistent Industrial Pollutants

Along with the development in industry, various persistent or non-degradable pollutants have appeared in the environment. Bisphenol A (BPA), a compound often used in the synthesis of various plastics, is detected in multiple types of water. As an analogue of estrogen, BPA affects the human genital system [80]. WH has been shown to quickly and efficiently adsorbed BPA and subsequently degrade it [56]. Perfluorootanoic acid (PFOA), an undegradable foaming agent, is commonly used in fire extinguishers and detergents. The PFOA residues in soil and water are believed to exert toxicity to humans via biological chain accumulation [81]. Mudumbi et al. (2014) determined the level of the PFOA in 11 aquatic plants sampled from wild rivers, and the results showed that PFOA in the WH roots reached the highest concentration at 38 ng/g [57]. In addition, WH was found to gather floating particles, such as plastics, thereby cleaning up the white pollution in water [58].

#### 2.1.7. Other Organic Pollutants

In addition to the above-mentioned organic pollutants, WH also showed decent remediation to other toxic organic pollution. For example, WH has been reported to absorb and degrade pentachlorophenol [61]. In addition, WH is highly tolerant to and capable of enriching the toxic cyanide. The EC50 was as high as 13 mg/L, while the max transfer rate of cyanide by WH fresh plants was 35 mg/h per kilogram [60]. WH was employed to repair cyanide pollution in the waste water of a steel factory, and the result showed 95% was removed within 3 d [82]. Moreover, WH has also been reported to repair the soil pollution of oil and palm oil plants containing palm oil plants [63,64].

### 2.2. Other Forms of WH Used in Pollution Remediation

Until now, the majority of research on the remediation effect of WH used intact living plants, while in vitro tissues showed comparable effects. Shabana and Mohamed (2005) reported that detached roots of WH efficiently absorbed As in wastewater [83]. Moreover, there are several other application forms of WH in remediation that should be paid special attention. WH ash has been reported to absorb phenol and heavy metal pollution from water, and the adsorption efficiency is higher at a low pH [30]. Biochar is another remarkable application form of WH and has shown substantial capacity for remediating a wide range of pollutants, including heavy metals, antibiotics, phenols, dyes, and so on [66,84,85,86]. Moreover, the WH derived-biochar could be made into composites with other materials to further improve the adsorption efficiency. For example, the adsorption efficiency of WH derived-biochar and graphene oxide-WH-polyvinyl alcohol composite towards azithromycin were 244.498 and 338.115 mg/g, respectively [54]. In addition to increased adsorption efficiency, the composites made with WH biochar and other materials have good performance in coping with multiple pollutants. For instance, the WH derived-biochar coated with Fe_3_O_4_ or Fe_2_O_3_ showed high sorption towards phosphate over pH 3–9 and appreciable sorption while antimonate, nitrate and sulfate co-exist [87]. The excellent absorption performance of this composite is attributed to its ferromagnetic property, which increases surface complexation with the hydroxyl via ligand exchange. Such adsorption increases were observed on WH-derived activated carbon embedded with Co nanoparticles toward methylene blue resulting from the enhanced adsorbing area [88]. In addition to the above-discussed straightforward absorption, WH-derived biochar also showed catalyst activity on the degradation of persistent contaminants. 4-nonylphenol (4-NP), a nonionic surfactant commonly used in industrial manufacturing, is a type of organic contaminant that is notoriously toxic and difficult to decompose and therefore threatens human health and ecological safety [59]. The WH-derived biochar promoted the calcium peroxide-mediated degradation of 4-NP in sediments [89]. The catalyst action of WH biochar is ascribed to its abundant electron-rich carbonyl groups and active sites provided by vacancies in the catalyst structure, as well as the graphitized carbon framework which acts as a medium for electron shuttling [89]. Furthermore, membrane is another new application form of WH in pollutant remediation. The addition of WH biomass into an aerobic hollow-fiber membrane increased the chemical oxygen demand and the removal efficiency of three antibiotics, erythromycin, sulfamethoxazole and tetracycline [55]. Furthermore, the novel cellulose-based ion-exchange resins made from WH presented high adsorption capacities for heavy metals Pb, Cd, Cu, and Ni [90]. The remarkable remediation of WH-based adsorbing materials could be governed by the chemisorption mechanisms in adsorbing various pollutants and by the effects of WH on shaping the microbiota in the reaction environment, which are likely to function vitally in degrading organic pollutants (as discussed Section 3.2.1) [54,55].

Compared to intact plants and detached tissues, the newly developed adsorbing materials made from WH, including biochar, composites, hollow-fiber membrane bioreactor and ion exchange resins exhibit a much higher adsorption efficiency. For example, the Cd adsorption efficiency of WH roots is 9.545 ± 0.006 mg/kg, while that of biochar reaches 46.8 g/kg [35,91]. Moreover, the adsorption efficiency of these new materials can be further elevated by adjusting the manufacturing processes and reaction parameters [66]. In addition, these materials can be used repeatedly, which means less environmental and economical costs. Moreover, these materials can avoid potential ecological problems brought by living plants, including threats to biodiversity and secondary pollution resulting from dead plants that are not harvested timely. Furthermore, the application scenarios of these materials are greatly extended. They can be used in conditions that are impossible for WH fresh plants to survive, in remediating air pollution, in purifying drinking water, and so on. Finally, these materials are much easier to manipulate, store, and transport, thereby further reducing costs.

Apart from the super capacity for remediating pollution, WH-derived materials also present great potential for green energy production, as discussed in the following parts (Section 2.3.1).

### 2.3. An active Participant in Coping with the Global Warming Crisis

Global warming has deteriorated into probably the most challenging and urgent environmental problem since the second industrial revolution. Intriguingly, WH is an active participant in curbing global warming by decreasing greenhouse gas (GHG) emissions via green energy generation, on one hand, and by functioning as a carbon sink, on the other (Figure 1).

#### 2.3.1. Generation of Green Energy

There have been a huge number of publications demonstrating and optimizing the productive processes of biofuels, such as ethanol, methanol, hydrogen (H_2_), biogas, and bio-oil, using WH as the feedstock [5,65,66]. The optimization includes different pre-treatment of raw materials, the employment of various bacteria and fungi strains in decomposing and fermentation, and adjusting reaction parameters, such as temperature, pH and so on [32,92]. By incubating with weak acid, white rot or brown rot fungus, the outer surface WH is more efficiently degraded into hemicellulose which is subsequently hydrolyzed into sugar, and eventually the conversion efficiency of ethanol reaches 0.192 g/g after adding yeast to promote fermentation [93]. By combining hot air oven, microwave, hot water bath and autoclave pretreatments, the production efficiency of methanol from WH reached up to 3039 mL g^−1^ [94]. As to the production of H_2_, by optimizing the reaction time, the concentration of sulfuric acid added, the reactor stirring speed, the hydrolyzed pH and other reaction conditions, the maximum preparation efficiency reaches 126.7 mmol L^−1^ [95]. The biogas production from WH petioles can be elevated to 170 L kg^−1^ by the pretreatment of 1-N-bityl-3-methyimidazolium chloride ([BMIM] CL)/dimethyl sulfoxide (DMSO) at 120 °C for 120 min, which increased cellulose content by 27.9% and reduced lignin content by 49.2% [96]. However, before preceding the energy production procedures, special attention should be paid to the storage of WH. Due to the high level of water content (up to 90%), improper storage would cause low energy conversion efficiency and mosquito breeding. Hence, calcium dioxide pretreatment is commonly used to preserve WH materials and to reduce the lignin contents in WH, which favors subsequent hydrolysis reactions [97]. Notably, the high content of water in fresh WH renders this plant great potential for preparing bio-oil through hydrothermal liquefaction. Furthermore, the addition of heavy metals Pb and Cr can increase the yield efficiency to 56% and 63%, respectively [98,99]. Moreover, the two metals mostly remain in the reaction residue instead of bio-oil, thereby ensuring the bio-oil meets the requirements of environmental sound [98,99]. In addition, because of the high level of free fatty acids (up to 65%), WH leaves were used to produce biodiesel using magnetic Fe_3_O_4_ nanoparticles (NPs) wrapped in Arabic glue at an efficiency of 87% [100].

In addition, several pioneer works shed light on the application of WH in super capacitors, a novel energy storage device, as well as in new batteries [67,68,69,70]. Shell et al. (2021) demonstrated that WH-derived biochar is a decent material in preparing supercapacitors for its high adsorption of Ni and therefore exerts electrochemical performance [67]. Likewise, Alzate et al. (2022) reported that a natural fiber-polyester electrode composited by polypyrrole and WH-derived polyester fabrics exhibited good performance in electroactivity, tensile strength, volumetric power and energy densities, suggesting that the composite is a promising super capacitor electrode [70]. Additionally, as an excellent heteroatomic carbon source, WH-based biochar can be used as electrocatalysts in catalyzing oxygen reactions in alkaline fuel cells or microbial fuel cells [68,69].

#### 2.3.2. Serving as a Carbon Sink

Because of its rapid growth and vegetative reproduction, WH is considered a potential plant for carbon capture. Interestingly, the increasing environmental temperature, resulting from the elevated atmospheric GHG, further favors the growth and spread of WH [3]. Compared with the photosynthetic efficiency of another aquatic plant, *Typha domingensis*, which is approximately 1.12 g m^−2^ day^−1^, the carbon dioxide (CO_2_) uptake of WH is relatively high, reaching 3.4–5.4 g m^−2^ day^−1^ [71,101] and is comparable to that of a C4 plant corn, which is 4.3–29.4 g m^−2^ day^−1^ [102]. Peixoto et al. (2016) showed that the net assimilation of WH in the studied lakes offset their CO_2_ emissions, leading to their function as a CO_2_ sink [71]. However, in addition to CO_2_, WH also emits another GHG, CH_4_, which presents a global warming potential of 34 times over CO_2_. Several studies have investigated the effect of WH on total GHG (CO_2_ and CH_4_) balance and shown that WH acts as a net GHG sink or acts neutral on GHG balance, mainly resulting from variations on CH_4_ emission [103,104]. These variations can be explained by differences in rooting and plant coverage, as lower water depth and higher density of plants favor more CH_4_ emissions [105]. Therefore, combined with plant harvest for other utilization, WH at moderate density is of great potential to act as a carbon sink.

## 3. Main Mechanisms Underlying WH Utilization in Environmental Sustainability

### 3.1. The Biological Characteristics of WH

#### 3.1.1. Rapid Growth and Reproduction Rate

WH is one of the fastest growing and reproducing plants, which normally propagates by producing clonal ramets. It is estimated that the growth rate of WH under normal conditions can reach 0.26 T dry substance d^−1^ (hm^2^)^−1^ [100,106], and the population can double within 10 days under favorable conditions [63]. The rapid growth is probably due to multiple photosynthesis- and transpiration-related protein analogues that exist in it [107]. Though the rapid rate of vegetative growth and propagation results in huge threats to biodiversity and other harm, this biological characteristic renders WH the greatest virtue in its utilization, which is readily sufficient in terms of feedstock availability. Additionally, the rapid growth and reproduction rate, resulting from high capacity in CO_2_ fixation, enables WH great potential in carbon capture.

#### 3.1.2. Unique Physical Properties and Phytochemical Composition

WH has well-developed fibrous roots, which not only maximize its contacting surface and absorbing efficiency but also its function in intercepting particles and larger contaminants, such as undissolved textile pollutants and floating plastic [58,66]. In addition, WH-derived materials have a relatively high surface area and pore size, which further enhances the great physical contact area and, thereby, promotes their remediation effect [32,66]. From the perspective of phytochemical composition, WH is a perfect plant for pollutant remediation. The phytochemical composition of WH consists of phenolic compounds (9.87%), flavonoids (10.49%), saponins (1.23%), terpenoids (11.72%), sterols (6.17%), alkaloids (7.4%), quinones and anthraquinones (2.41%), phenalene derivatives (15.43%), carbohydrates (6.79%), organic acid (9.25%), and other compounds (19.13%) [13]. The rich contents of antioxidants, such as phenolic compounds, sterols, terpenoids and other antioxidants, render WH substantial radical scavenging activity and therefore protect WH’s growth in polluted toxic water [108,109,110]. For example, it can grow in water where the concentration of mixed heavy metals is as high as 3 mg L^−1^ [111]. In response to toxic pollutant stimuli, the contents of antioxidants significantly increased [107,112]. Likewise, the levels of antioxidative enzymes, such as superoxide dismutase, peroxidase, catalase, ascorbate peroxidase and ascorbic acid peroxidase, were also elevated by pollutant stress [107,113,114]. Moreover, since heavy metals often harm plant photosynthesis and transpiration systems, WH has another mechanism to ensure its growth by having more photosynthesis- and transpiration-related protein analogues [107]. In addition to the growth protection exerted by antioxidants, antioxidative enzymes and some other proteins, some other chemicals facilitate the great chemisorption of WH towards pollutants. Ca-oxalate crystals and humic acid are two classes of verified metal-binding chemicals [115,116]. In addition, the high level of proteins and carbohydrates that exist in WH provide extensive chelating sites for metal irons and, therefore, enable WH to achieve good performance in heavy metal absorption, resulting from their abundant functional groups, such as carbonyl, carboxyl, hydroxyl, and so on [117]. Interestingly, the chemisorption of WH can be enhanced by metal ions, such as Cu^2+^, which forms a strong metal bridge on the root surface of WH under weak acidity conditions [77,78]. In addition to the above discussed chemisorption, some bioactive compounds in WH were supposed to work in the assimilation of organic pollutants, because the detached roots and the crude extracts of WH roots showed comparable adsorption [62]. This may partially account for the degradation of many organic pollutants inside WH plants. Meanwhile, WH-related micro-organisms also play a role in organic pollutant degradation, which will be discussed in Section 3.2.1. Furthermore, the abundant functional groups render WH-derived biochar catalyst activity on the calcium peroxide-mediated degradation of 4-NP. The abundant electron-rich carbonyl groups and active sites are provided by vacancies in the catalyst structure, as well as the graphitized carbon framework which acts as a medium for electron shuttling [89].

In addition, the unique chemical composition enables WH to be an excellent raw material in clean fuel production. WH contains 33.1–46.5% carbon, 4.4–6.6% hydrogen and a high total thermal value (13.1–18.4 mJ/kg) [118,119]. Compared with other plants that are often used in bioenergy production, such as sugarcane bagasse and wheat straw, WH contains comparable levels of hemicellulose (34.1%) and cellulose (24.5%) but less lignin (8.6%) [92]. Lignin, compared with hemicellulose and cellulose, is much more difficult to convert into energy. Therefore, WH is a favorable source for producing green fuels [92]. As a raw material in energy production, another advantage of WH is its C/N ratio (approximately 20:1), which favors microbial decomposition in processing procedures [25].

### 3.2. Assistance from Micro-Organisms

#### 3.2.1. Effects of Rhizosphere and Endophytic Micro-Organisms on WH Phytoremediation

In addition to plants, micro-organisms are active participants in pollution remediation by promoting plant growth in a polluted environment, on one hand and by degrading or assimilating pollutants, on the other. Therefore, phytoremediation by plants is often combined with microbial remediation to achieve higher efficiency. Likewise, multifunctional micro-organisms are indispensable to the excellent phytoremediation presented by WH (Figure 1). Antibiotics have been shown to be negatively correlated with the growth of WH, for instance, the oxytetracycline treatment significantly decreased the wet weight of WH [120,121]. Additionally, inoculation with *Bacillus subtilis*, *B. mucilaginosus*, or *Aspergillus niger* in WH roots promoted its biomass by increasing the maximum photochemical efficiency [122]. Moreover, the exogenous addition of *Pseudomnas aeruginosa*, *Escherichia coli*, *Klebsiella ozanae*, *K. edwardriella* and *B. subtilis* ensured the growth of WH in toxic raw sewage, while non-inoculated plants could barely survive [123]. These studies uncovered the growth-promoting effect on WH of micro-organisms and partially explained the growth of WH in toxic polluted water. On the other hand, micro-organisms play a key role in the adsorption and/or degradation of WH toward various pollutants. For example, antibiotic treatments significantly reduce the remediation efficiency of Se, Cu and Pb by WH [121,124]. In addition, six bacterial isolates of which 16S rRNA sequences showed high similarity to *P. diminuta*, *Brevundimonas diminuta*, *Nitrobacteria irancium*, *Ochrobactrum anthropi*, and *B. cereus*, respectively, were isolated from the rhizosphere of WH [125]. These bacteria promoted the adsorption efficiency of WH toward heavy metals (Mn, Zn, Cr) to different extents. Moreover, the microbial species in WH roots were investigated, and the results suggested that they play key roles in the remediation of nitrogen and metals [124]. In particular, *T. thiooxidans*, *T. ferrooxidans*, *Azotobacter* and *A. niger* are responsible for the absorption of metals tested in the assays (Zn, Cd, Ni, Fe, Cu, Sb, Sn and Cr), while *Azotobacter* accounts for nitrogen fixation, which could perfectly explain the growth promotion of WH by micro-organisms. Additionally, inoculation with *B. subtilis*, *B. mucilaginosus*, or *A. niger* in the WH roots significantly increased its U enrichment capacity [122]. Furthermore, the degradation efficiency of various pesticides, such as omethoate, ethion, dicofol, cyhalothrin, chlorpyrifos, mesotrione and fomesafen, is positively correlated with WH endophytic micro-organisms [76,84,126]. Likewise, two isolated endophytic *Bacillus* strains obviously promoted the adsorption of multiple organic and inorganic pollutants by WH [127]. Until now, the majority of micro-organisms that have been shown to be involved in WH remediation are rhizospheres and endophytic bacteria, but the effects of fungi are non-neglectable [128].

#### 3.2.2. Micro-Organisms Play Vital Roles in Biofuel Production

Anaerobic digestion (AD) whereby carbohydrate polymers are converted into monomers is a key process in producing biogas, biomethanol, bioethanol and biohydrogen. AD efficiency, which determines the final biofuel yields, heavily relies on the activities of micro-organisms functioning in hydrolysis, fermentation, acidogenesis and methanogenisis, including bacteria, archaea and fungi. Archaea predominantly work in methanogenisis, while bacteria and fungi often contribute to other processes [129,130,131]. Activities of the microbial consortium depend on the species and the amount of initial inoculum, and their growth during the process. The amount of initial inoculum is indicated by the Food/Microbe (F/M) ratio. It has been suggested that biofuel production reached the highest when the F/M ratio was 10.01 [132]. The growth of the microbial community is affected by reaction parameters including pH, temperature, and nutrient balance, such as the C/N ratio. Thus, the microbiome changed along with the alternations of these parameters during AD [133,134]. To further enhance sustainability, WH is often co-digested with other plant biomass, animal dung and other wastes [135,136]. The co-digestion could increase biofuel yield, most likely because the mixed substance contains nutrients at a more suitable C/N ratio and consequentially provides a more suitable pH and temperature, thereby favoring the growth of microbial consortiums [135,137].

## 4. Conclusions

As discussed above and shown in Table 1 and Figure 1, WH influences environmental sustainability through two aspects: pollution remediation and curbing global warming. In pollutant remediation, WH could be used to remove not only the well-known traditional pollutants, including metals, other inorganic elements, dyes, pesticides, and other organic toxic pollutants, but also newly developed pharmaceutical and industrial residues. Moreover, an array of WH-derived materials used in pollutant remediation should be highlighted, because of their superiority in terms of adsorption efficiency, environmental effect, application cost and range. In coping with the global warming crisis, WH participates in two aspects: green energy generation and carbon capture. By optimizing reaction parameters in bio-fuels generation and extending its application in producing super-capacitors and new batteries, WH showed great potential in green energy generation. In contrast with other abovementioned WH-related topics, WH functioning in carbon capture has gained much less attention so far. However, based on the current data and the prediction of its growth and spread in the future, we can conclude that WH is a promising carbon sink. The multi-function of WH in maintaining environmental sustainability is attributed to its biological characteristics and related micro-organisms (Figure 1). Specifically, micro-organisms play vital roles in green energy production by dominating anaerobic digestion and in pollution remediation by promoting plant growth, pollutant adsorption and/or degradation.

Overall, this review summarizes various utilizations of WH in environmental sustainability, especially recent significant progress, and deciphers the underlying mechanisms. Although considerable progress has been achieved, related studies mostly remain at the phenotypic or physiological level, while molecular information is barely discussed and requires much more effort in the future.

## Figures and Tables

**Figure 1 ijerph-19-16725-f001:**
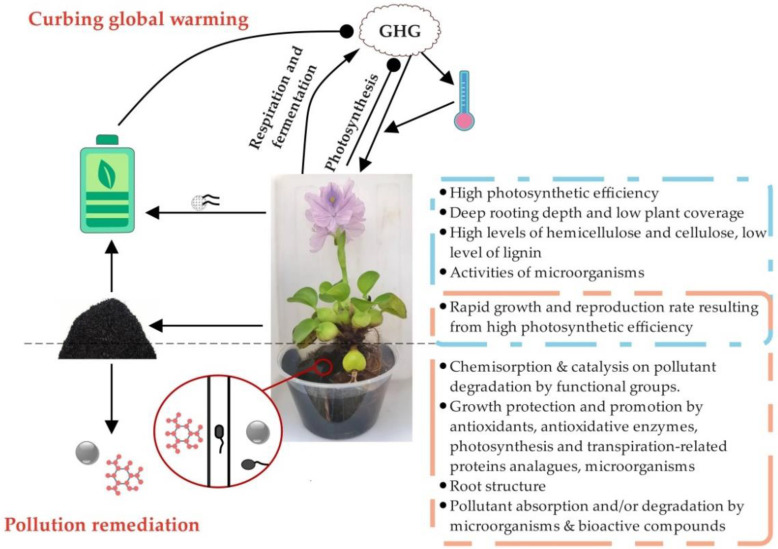
A schematic model deciphering the modes of action of water hyacinth in maintaining environmental sustainability. GHG: greenhouse gas. WH affects environmental sustainability by curbing pollution remediation and global warming. On one hand, WH adsorb and/or degrade a wide range of contaminants in the forms of living plants and other WH-derived materials. On the other hand, GHG favors WH growth not only by boosting photosynthesis efficiency, but also by increasing environmental temperature. In turn, WH negatively affects the environmental GHG concentration by CO_2_ assimilation on one hand and by generating green energy in the forms of plants and WH-derived materials on the other. Meanwhile, WH emits variable amounts of CO_2_ and CH_4_, via respiration and fermentation, respectively. Additionally, WH impedes the GHG increase by its application in green energy generation. The texts in the blue rectangle and in the orange rectangle illustrate the main mechanisms underlying utilization of WH in curbing global warming and pollution remediation, respectively. The lines with arrow ends represent a positive effect, while the lines with circle ends represent a negative effect.

**Table 1 ijerph-19-16725-t001:** Utilization of water hyacinth in maintaining environmental sustainability.

Function	Category	Details	Utilization Forms	References
Pollutant remediation	Metals	Ag, Al, Au, Cd, Co, Cr, Cu, Eu, Fe, Gd, Hg, Mn, Ni, Pb, Sb, Sn, Sr, Ti, U, V, Zn	Living plants, detached tissues, biochar, nanoparticle composites, membrane, resins.	[5,30,31,32,33]
	Other inorganic pollutants	As, N, P, S, Se		[5]
	Dyes	Acid yellow 17, alkaline auramine O, azharanth dye, BF-4B red active dye, black B, BreActive magenta B, C. I. acid blue 19, C. I. acid blue 25, C. I. acid blue 80, C. I. acid green 27, C. I. acid red 1, C. I. acid red 40, CI direct blue 201, cibacron blue FR, cibanone gold yellow RK, congo red, cotton blue B2G, cotton red B2G, cotton yellow 2RFL, cresol red, crystal violet, indosol dark-blue GL, malachite green, methylene blue, methyl red, methyl orange, moxilon blue GRL, phenol red, reactive black 5, reactive blue 21, reactive turquoise blue, red RB, rhodamine B, rose bengal, phenol red vat green FFB, and xylenol orange.		[2,37,38,39,40,41,42]
	Pesticides	BHC, β-BHC, δ-BHC, γ-BHC, β-endosulfan, heptachlor epoxide, dichlorodiphenyltrichloroethane, dicofol, cyhalothrin, organophosphorus pesticides, chlorpyrifos, triazine, diazinon, ethoprophos, malathion, methylparathion, omethoate, ethion, diphenamid, mesotrione and fomesafen.		[43,44,45,46,47,48]
	Pharmaceutical residues	Azithromycin, ciprofloxacin, erythromycin, sulfinirazine, tetracycline, naproxen, ibuprofen, and diclofenac, fenoprofen, emtricitabine, tenofovir disoproxil and efavirenz.		[49,50,51,52,53,54,55]
	Newly developed persistent industrial pollutants	BPA, PFOA, plastics, 4-NP.		[56,57,58,59]
	Other organic pollutants	Phenols, cyanide, oil.		[60,61,62,63,64]
Curbing global warming	Green energy	Biofuels: ethanol, methanol, H_2_, biogas, and bio-oil.	Intact plants or detached tissues	[5,65,66]
		Supercapacitors, alkaline fuel cells and microbial fuel cells	Biochar	[67,68,69,70]
	Carbon sink	Carbon capture.	Living plants	[71]

BHC: benzene hexachloride; BPA: bisphenol A; PFOA: perfluorootanoic acid; 4-NP: 4-nonylphenol.

## Data Availability

Not applicable.

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
