# Peer review of "The Multifaceted Function of Water Hyacinth in Maintaining Environmental Sustainability and the Underlying Mechanisms: A Mini Review"

_ijerph, 2022, doi:10.3390/ijerph192416725_

Round 1

Reviewer 1 Report

Water hyacinth (WH) is a noxious invasive aquatic weed, causing severe ecological and economic losses. Interestingly, resource utilization seems to be a sustainable method to control it. The manuscript entitled “The multifaceted function of water hyacinth in maintaining environmental sustainability and the underlying mechanisms: A mini review” by Xu et al. summarized WH utilization in pollutant remediation, green energy production, and curbing global warming. This manuscript provides a valuable reference for readers to understand the latest research progress on these topics. More importantly, this manuscript discussed the main mechanisms underlying these utilizations, which are rarely seen in other similar reviews. Yet, I have several suggestions to improve the manuscript:

1. The authors need to delete some unnecessary information to keep the manuscript as concise as possible. For example, on page 4, lines 27-31, it is not necessary to introduce too much about the significance and effects of pesticides.

2. Section 4.1.2 should be corrected into 3.1.2. 

3. The authors should re-structure sections 3.1.3 and 3.1.2 to enable this part to be more comprehensive. Try to integrate these two sections and make the logic clearer. 

4. Add reference to “As, N, P, S, Se, Si” and “Biofuels: ethanol, methanol, H2, biogas, and bio-oil.” in Table 1. 

Author Response

Thank you very much for your comments and suggestions. We have addressed all your suggestions and made corresponding revisions, please check the revised manuscript.

Reviewer 2 Report

It is a relevant and well-written mini-review on water hyacinth as a phytorremediation solution among other potential qualities like carbon sequestration, while being a widespread invasive species that represents a real problem in some areas and causes economical and ecological losses.
Very interesting.

Author Response

Thank you very much for your comments. We are glad that you enjoyed our work.

Reviewer 3 Report

For the Authors
Congratulation for the good work. Some notes:
line 52 correct with this [16-19]
no space between the lines 86-87
spaces between the lines 96-97, 105-106, 122-123, 135-136, 158-159, 171-172, 242-243, 282-283, 310-311, 396-397

Author Response

Thank you very much for your comments. We have made revisions according to your suggestions.